# Exploratory Factor Analysis of a French Adapted Version of the Substance Abuse Attitude Survey among Medical Students in Belgium

**DOI:** 10.3390/ijerph20075356

**Published:** 2023-03-31

**Authors:** Lou Richelle, Michèle Dramaix-Wilmet, Nadine Kacenelenbogen, Charles Kornreich

**Affiliations:** 1Unité de Recherche en Soins Primaires ULB, Faculty of Medicine, Université Libre de Bruxelles, 1070 Bruxelles, Belgium; 2Département de Médecine Générale, Faculty of Medicine, Université Libre de Bruxelles, 1070 Bruxelles, Belgium; 3Département d’Epidémiologie et de Biostatistiques, School of Public Health, Université Libre de Bruxelles, 1070 Bruxelles, Belgium; 4Laboratoire de Psychologie Médicale et d’Addictologie, Faculty of Medicine, Université Libre de Bruxelles, 1020 Bruxelles, Belgium

**Keywords:** substance use disorder, attitudes, medical students, education, assessment, stigma

## Abstract

To evaluate the impact of a new Substance Use Disorder (SUD) education program on medical students’ attitudes, we selected the Substance Abuse Attitude Survey (SAAS) questionnaire, which we adapted to our curriculum and cultural context. To validate this adapted version, we conducted an exploratory factor analysis following the administration of our 29-item bSAAS questionnaire to 657 medical students in Belgium (response rate: 71.1%). Twenty-three items correlated to three factors; namely, “Stereotypes and moralism”, “Treatment optimism” and “Specialized treatment” were retained (70% of total variance explained, Cronbach’s alpha = 0.80) and constituted the new questionnaire called beSAAS. The factor “Specialized treatment” stood out from previous studies, which could be explained by our target population and the impact of the formal, informal and hidden curricula in medical education. This study was able to highlight certain factors influencing stereotypical representations such as age, gender, origin, personal or professional experience with substance use. Our study allowed us to retain the beSAAS as a good questionnaire to evaluate SUD stigma and highlighted interesting findings to improve SUD training in medicine. Further studies are needed to complete its validity and reliability.

## 1. Introduction

As part of the implementation of a new optional substance use disorder (SUD) training program for final year medical students, we wanted to assess its impact on students in terms of skills development (pre- and post-test evaluation). To evaluate the impact of the educational program on trainees’ representations and attitudinal skills development, we looked for a questionnaire to assess their attitudes towards SUD adapted to our criteria. Various instruments were found in the literature in French and English, but they had several limitations for our study. The first was that most of them focused on either illicit drugs or alcohol or perceptions about professional attitudes rather than about substance use itself (e.g., Drug and Drug Problems Perception Questionnaire and Alcohol and Alcohol Problems Perception Questionnaire [1,2].) We wanted to be able to assess both. Others were not culturally transferable to our context (e.g., Attitudes to Mental Illness Questionnaire [3]), some were limited in the dimensions they explored (e.g., The Addiction Beliefs Scale [4], Short Understanding of Substance Abuse Scale [5], Attitudes and Opinions Survey [6]), or were difficult to use on a large scale given their qualitative or vignette analysis approach [7]. Many also had limited «validity» and «reliability», not having been used beyond the study itself. On this basis, we selected the Substance Abuse Attitude Survey (SAAS) developed by Chappel et al. in 1985 [8]. Even though this questionnaire was developed more than thirty years ago, it was still relevant in our context. Indeed, it was designed to evaluate educational programs in the framework of initial or continuing education [9] and allows for the exploration of five dimensions (also called factors). These are “Permissiveness” (implies accepting substance use within a continuum of normal human behavior), “Treatment intervention” (relates to an individual’s orientation toward perceiving substance use/misuse in the context of treatment and intervention), “Non-stereotypism” (relates to a person’s non-reliance on popular societal stereotypes of substance use and substance users), “Treatment optimism” (relates to an optimistic perception of treatment and the possibility of a successful outcome) and “Non-moralism” (linked to an individual’s absence or avoidance of a moralistic perspective when considering substance use and substance users). This questionnaire has been validated and used in many previous studies [10,11,12,13]. In our study, we selected its short version, the “Brief SAAS” [14], to maximize its completion since in our program it was administered in parallel with a knowledge test. Given that an adaptation to our cultural and societal context was necessary, we named this new questionnaire the “bSAAS” (See Table A1 in Appendix A). We added questions about personal characteristics identified in the literature as influential in terms of SUD representations such as gender, age, origin, experiences related with substance use and choice of specialty [15].

The first objective of this study was to carry out an exploratory factor analysis of our bSAAS questionnaire in view of its adaptation to ensure its good internal consistency and structure. The second was to identify whether students’ representations were influenced by certain socio-demographic characteristics.

## 2. Materials and Methods

### 2.1. Questionnaire

The SAAS questionnaire validated widely by the literature consists of 50 items. For our study, we preferred its brief version, the Brief SAAS, which is used by the Yale School of Medicine to evaluate their SBIRT (Screening, Brief Intervention and Referral to Treatment) training programs and that is limited to 25 items. We adapted it to the Belgian context through consultation with various experts (including 3 general practitioners working in addiction medicine, 2 medical researchers, a SUD expert psychiatrist, 2 health sociologists and a professor of psychology). We removed the questions about marijuana experimentation among young people and Alcoholics Anonymous, which are less present in Belgium than in the U.S. We added a question on paramedical professionals, who are much more involved than para-professional counsellors in our context. After consensus, we split the questions on alcohol and drugs to be able to assess whether there were different attitudes according to the substances consumed. This resulted in a questionnaire with 29 items (Table A1 in Appendix A). We assumed that perceptions of legal and illegal substances might differ. Indeed, a common perception of the Belgian population is that the term “drugs” refers de facto to illegal drugs. This difference has been highlighted in studies conducted among Belgian doctors [16,17]. Each item was coded according to a Likert scale ranging from 1 (strongly disagree) to 5 (strongly agree). To ensure the cross-cultural validity of the questionnaire, we used bilateral translation by two certified translators and pretested the questionnaire with trainees and medical doctors from different professional backgrounds and with lay people. 

In order to characterize our population and to better understand the factors favoring the students’ representations, we included several socio-demographic data (gender, age, origin); data related to the personal and professional background linked to substance use; data related to the type of professional orientation (choice of specialty); as well as data related to the perception of their own health (we wanted to evaluate whether this influenced the way in which people with SUD are perceived). This questionnaire, which was called “bSAAS” by our research team, has already been used in a previous study to evaluate the attitudes of medical students towards substance use in pregnancy and showed a good internal consistency (Cronbach’s alpha = 0.77) [15].

### 2.2. Data Collection

The questionnaire was presented to 923 final-year medical students of three consecutive years (2019, 2020 and 2021). It was administered face-to-face to students in 2019 and online in 2020 and 2021 given the context of the SARS-CoV-2 crisis. A total of 657 students completed the questionnaire with an average response rate of 71.1%. Eighty students completed the questionnaire at the time of enrollment in the theoretical addictology course. For factor analysis, records with missing items were not considered. One or two items were missing in thirty-two (4.9%) and three to eleven items in ten (1.5%) questionnaires. On this basis, 615 (93.6%) questionnaires were retained.

### 2.3. Statistical Methodology

First, we analyzed the correlations between the 29 items and performed Bartlett’s test of sphericity to ensure that factor analysis was appropriate. The correlation coefficients were qualified as weak (≤0.30 in absolute value), moderate (between 0.31 and 0.50), good (0.51–0.74) and excellent (>0.75). The factor analysis was carried out in 3 steps and we decided to retain the factors with eigenvalues > 1 and with at least 3 items with a weighting >|0.40|, while keeping a sufficiently high % of variance explained (75%) [18,19]. Factor scores were obtained for each factor of our final factor analysis, retaining for each factor only those items with a weighting >|0.30| and taking the mean of these items. Five items were found in two factors; in this case, the item was assigned to the factor with the highest loading or, if the loadings were nearly equal, to the one that most closely matched the content of the factor. The mean scores were analyzed according to subjects’ characteristics using Student’s *t*-tests or one-way Analysis of Variance (ANOVA). Where appropriate, ANOVA was followed by multiple comparison tests with Bonferroni’s correction or a linear trend test. Cronbach’s alpha was calculated to measure the consistency of the 23 items included in the analysis as well as the consistency of the items selected for each factor separately. All analyses were performed with STATA SE v16.1 software.

## 3. Results

The examination of the correlation matrix showed that several items were correlated with correlation coefficients ranging from r = 0.32 to r = 85. The Bartlett test was significant (*p* < 0.001). The Kaiser–Mayer–Olkin (KMO) measure was 0.71 and our sample including 615 subjects was large enough to perform factor analyses.

Five items and one item were not retained after the first and second factor analyses, respectively (see Table A3 in Appendix B). The final factor analysis included 23 items and three factors were retained. This led to a new questionnaire which we called “beSASS”. Cronbach’s alpha was 0.80 for all 23 items. Considering the items with a weighting >|0.30|, the Cronbach’s alphas were 0.8, 0.70 and 0.59, respectively, for the 10 items of Factor I, the 9 items of Factor II and the 4 items of Factor III. We kept the same factor name for Factor I as mentioned by Jenkins et al. [10], i.e., “Stereotypes and moralism”, because the items correlated with this factor also reflected value judgments. As in that study conducted on secondary school students, which retained only three factors (stereotypes and moralism, treatment and permissiveness), the students in our study did not seem to differentiate between stereotype and moralism items. 

The name “Treatment optimism” for Factor II, found in Chappel et al., 1985, and called “Treatment” in Jenkins et al., 1990, has also been retained in our study. As in the first study, we find here negative correlations in relation to this factor since we have kept these items stricto senso (not recoded in the other direction). This also enabled us to better define this factor with items correlated negatively and therefore opposed to optimism, such as the item “An alcoholic who has had several relapses is unlikely to be treated”.

Factor III differed from the baseline studies [8,10] and seemed to be clearly marked by the specialized domain, with items correlated to hospital and specialist management in the field. We therefore named it “Specialized treatment”. These different factors and the corresponding items are shown in Table 1.

Concerning the three dimensions or factors retained in our final analysis: “Stereotypes and moralism”, “Treatment optimism” and “Specialized treatment”, we separated them into two axes based on the three major learning domains conceptualized by Benjamin Bloom in 1956 (cognitive, affective and psychomotor) and further developed thereafter [20,21]. On one hand, we identified a psycho-affective axis, based on moral judgments, and on the other, a cognitive axis, linked to the perception of health care skills. The latter includes Factor II, which evaluates a dimension linked to results (the perception of the effectiveness of the treatment, “successful outcome”) and Factor III concerning the perception of the needed resources (material and human resources). The correlations between the scores derived from these factors (r = −0.32, 0.23 and −0.17 for score 1 and score 2, score 1 and score 3 and score 2 and score 3, respectively) were weak to moderate, which is an additional argument for maintaining a three-dimensional questionnaire. The score “Stereotypes and moralism” was positively correlated with the “Specialized treatment” score and negatively correlated with the “Treatment optimism” score. The score “Specialized treatment” was positively correlated with the “Stereotypes and moralism” score and negatively correlated with the “Treatment optimism” score (see Figure 1).

It is to be noted that the items dealing with the same content for drugs and alcohol had good or excellent correlations (r = 0.55–0.85).

Secondly, we analyzed the factor scores according to the different socio-demographic data collected. There was a significant increase in the mean score for “Stereotypes and moralism” (Factor I) with age. This mean score was also statistically significantly higher for male respondents (*p* = 0.037). The differences in the average score for “stereotypes and moralism” according to “subject-parent” origin were significant (*p* = 0.005); among people of non-European origin, we observed a higher average score than that of people of Belgian or Belgian–European origin. Subjects who did not consume any substance had a higher “Stereotypes and moralism” mean score than people who had consumed (only cannabis or multiple substances) but the differences were at the limit of statistical significance (*p* = 0.065). Conversely, the average score of those who had ever used cannabis was low. This was also the case for those who had been in contact with patients with SUD in specific centers (e.g., addiction centers, harm reduction centers, prisons, etc.) or who had enrolled in the optional SUD training, who differed significantly from the other respondents in having very low average scores on “Stereotypes and moralism”. There was no statistically significant difference in the average factorial I scores according to the perception of one’s own health and SUD in the environment. The same was true for the choice of specialty, although students who wanted to go into gynecology had a lower average score. There was no statistically significant difference in the mother’s level of education either, although for this last variable the average score was higher for those whose mother’s level of education was low. 

For Factor II, “Treatment optimism”, the average score remained close to the mean of the total sample (2.94) across the different socio-demographic characteristics. This means that there was a tendency, independent of subject characteristics, to be optimistic towards treatment and possible interventions. However, the 23 respondents who had no contact with people with SUD at work showed a clear tendency towards low optimism about treatment (average score lower than the others, NS). 

In relation to Factor III, which assessed attitudes towards the specialized treatment of SUD, male respondents had a significantly lower mean score (*p* = 0.035). People whose origin was outside Europe had a significantly higher mean score than Belgian–Europeans (*p* = 0.019) or Europeans (*p* = 0.049). We also found a tendency for people with no contact with SUD patients, or with contact in hospitals, to be in favor of “Specialized treatment” (NS). On the other hand, those who went to general practice were not in favor of “Specialized treatment”. The same was true for respondents enrolled in addiction training (*p* = 0.031). Detailed results can be found in Table A4 (see Appendix B) and Figure A1 (see Appendix C). 

To investigate if the SARS-CoV-2 crisis could have impacted students’ representations, we compared factors and variables according to the three student cohorts (2019, 2020 and 2021). The results did not show any relevant significant difference except for a slight decrease in the average score of Factor I over the years (from 2.14 to 2.02; *p* = 0.05).

## 4. Discussion

Our study, through the adaptation of a short version of the Substance Abuse Attitude Survey questionnaire administered to medical students in Belgium, allowed us to select a 23-item questionnaire with good face validity and content validity (via expert panel and pre-test), construct validity (via exploratory factor analysis, convergent correlations between alcohol and drug items and discriminatory correlations between factor scores) and very good internal consistency (Cronbach’s alpha = 0.80). That allowed us also to highlight interesting results concerning influencing factors of medical students’ attitudes regarding SUD. The three dimensions selected: “Stereotypes and moralism”, “Treatment optimism” and “Specialized treatment”, which we divided into two axes (affective and cognitive), are useful for exploring the learning objectives pursued in our educational system. These results can also guide us in the methodological choices to be made to achieve these objectives. 

We can see some differences with the reference studies [8,10]. As the latter pointed out, the factor structure logically changes according to the characteristics of the people completing it. Our target population and questionnaire were somewhat different from those of the first study, conducted on different profiles of already experienced professionals (non-clinicians, clinicians not in the field and clinicians in the field) and those of the second study conducted on college students (first university degree). This may partly explain the difference in the factors observed. 

In our study, given the circumstances (SARS-CoV-2 crisis) and the survey having been carried out partly online, the response rate (71.1%) was good. We found an over-representation of women, which corresponds to the feminization of medicine over the last few years and is in line with the latest gender report from our university [22], which reports 62% female students in medicine. We also see that 72.7% of students have a mother with a high level of education. These attitudes therefore reflect a selected population of future caregivers who are very likely to differ, also because of the number of years between these studies, from the populations studied in previous articles and from the general population at the socio-demographic level (in terms of gender imbalance and education level). 

With regard to the factors, it is firstly interesting to note that the dominant factor in our study (explaining 40.3% of the variance) was, as with that of the college students, the factor relating to “Stereotypes and moralism”. This differs from the case of Chappel et al., 1985, where it was the factor relating to permissiveness. The predominance of this affective dimension in a medical student cohort can be explained in different ways. Firstly, we hypothesize that, as the SUD topic is little taught in the formal curriculum, its representation is at this stage still partly based on representations conveyed by society or by students’ communities. These representations are probably also influenced by their internship experience, where the impact of role modeling (defined here as “*a teaching by example and influencing students in an unintentional, unaware, informal and episodic manner*”) [23] is documented in students [23,24]. Students may integrate both the positive and negative behaviors of their role models and mentors, perhaps more so in the context of choosing a career path, where they are looking to identify models. The informal and hidden curricula play a key role in the transmission of attitudes and values [24]. This is consistent with the study of Kidd et al. in 2020 [25], which justified the fact that students tended to adopt more negative attitudes towards SUD people over time by the potential impact of the “hidden curriculum” through which students internalize the “negative” attitudes of their supervisors. The hidden curriculum includes “*a set of values, behavioral norms, attitudes, skills, and knowledge that medical students learn implicitly*” [26]. These attitudes are influenced by social and institutional stigmas that lead to suboptimal care including less engagement and empathy by health professionals [27]. By perceiving these stigmas, and by a phenomenon of self-stigma, SUD people may seek less help and be less involved in treatment, which can lead them to exclude themselves from the healthcare system [28,29].

In our study, the (non) place of contact seemed to play an important role in stereotypes and moralism, the latter being particularly marked if there was no contact or if it took place only in the hospital system. Conversely, scores were low if students had had multiple contacts or had experience of specific settings for patients with SUD such as SUD centers. This demonstrates the importance of varied and supervised experiences to reduce stereotypes. The fact that some students have never used substances on their own has also encouraged these stereotypes [15,27,30]. 

The difference between the factors emerging from our study and our reference articles can also be explained by the fact that in our final analysis we no longer had any of the items correlated with the “Permissiveness” factor initially present in the study by Chappel et al. Some of those items were not included in the “Brief SAAS”, some of which differed from the initial questionnaire, and other items were removed from the “bSAAS” because they were not adapted to our context, such as experimental marijuana use among young people, or were not retained during the second or third stage of the factor analysis leading to the “beSAAS questionnaire” (See Table A2 in Appendix A). 

In relation to the third factor, which we decided to name “Specialized treatment”, it was well characterized in our study by the four items correlated with it (referring to care by specialists and in hospital in cases of alcohol or drug use disorders) and thus seems to be better defined here rather than in previous studies where they were associated with the notions of stereotypes and moralism among other items. Nevertheless, it is stereotypical to believe that the complexity of the situations should be dealt with by the second line of care. Recent studies on opiate use disorders show, in contrast, that the primary care setting is the most appropriate [31,32], although this is more controversial for alcohol, even if its role of early detection and intervention is essential [33]. The fact that in our study this emerges as a factor on its own, in contrast to the two studies mentioned above, is indicative of our target population. Here, we can also see the impact of the “hidden curriculum” on medical students, as mentioned by Sc. Mahood in 2011 [34]. Indeed, in the medical curriculum of our Faculty, the importance and primacy of specialties are regularly emphasized outside the formal curriculum. The training is mainly hospital-centered as well. The fact that general medicine is not sufficiently valued in the learning process is certainly internalized by the students, especially a few months before the final exams when it comes to the selection of specialism, and may influence these stereotypical representations (these two scores being positively correlated in our study, albeit moderately). 

As for the results of the factor scores, we can observe that the characteristics of the subjects with higher scores for stereotypes and moralism are similar to those who were in favor of punishing substance use during pregnancy (alcohol/drugs) in our previous study [15], these two items being the most «weighted» in this first factor.

Lastly, it is interesting to note that the students who enrolled in SUD training scored lower on stereotypes and moralism compared to those who did not decide to attend. Our study highlighted also an interesting result for our SUD training: specific contact with SUD people was associated with a lower score in stereotypes and moralism. Furthermore, there was not a statistically significant difference in terms of optimism about treatment between the two groups, which was a tendency to be in favor. These findings lead us to the conclusion that we should involve people in recovery and peer helpers in this optional training to contribute to lowering stigma and to help improve the perception of treatment outcomes through the sharing of positive experiences. We should also focus our teaching on recovery approach and long-term support for this chronic disease (“care vision”) rather than focusing on curing the individual (“cure vision”) as is commonly taught in medicine. Even if we could observe a slight improvement in stereotypes and moralism from year to year, which could be explained by more awareness among medical students of mental health and substance use disorders in a context of overall vulnerability, there is still work to do. This study also reveals the interest in SUD and necessity of making a SUD training program compulsory for all medical students, in addition to the basic SUD education, to improve attitudes, access and quality of care for people with SUD [27,35,36]. It should be noted, however, that the fact that the people taking part in the training were generally not in favor of specialized treatment for this group can be explained by the way in which this training is offered by the Department of General Medicine, and the fact that students that are heading for general medicine are the more likely to take part. 

Although this study made it possible to validate a questionnaire for evaluating representations in French that could be used to evaluate educational systems, it does have certain limitations. First of all, we obtained an acceptable average response rate of 71.1%. This result was unfortunately impacted by a low response rate in 2020 (47.3%) due to the first wave of the pandemic and students’ other concerns, which limited the representativity of the sample especially for this cohort. For the rest, the response rates were significantly higher, with 82% and 79.3% of respondents for 2019 and 2021, respectively.

Otherwise, we were not able to ensure its external validity by measuring its correlation with the basic instrument or other related instruments. Nor have we assessed its reliability in terms of repeatability (test-retest) and reproducibility on other profiles. It would also be useful to conduct a confirmatory factor analysis on the basis of our new 23-item questionnaire (beSAAS) to validate its construct. Further studies could be conducted in the future to do so. It is also a questionnaire with limited items filled by the students themselves, which has its limitations in assessing students’ attitudes in practice. Indeed, it is not reflective of all the potential attitudes related with SUD and could be completed by patients and SUD trainers in the practice setting to be more accurate and comprehensive. To have better knowledge of the topic, we could also complete this study by semi-structured interviews or health simulation training to be able to analyze student attitude, discourse and implicit bias in more depth. As this questionnaire was administered to medical students, which means future caregivers, we suspect that the results suffer from social desirability bias. This could result in the reporting of more positive attitudes than those experienced in reality. 

## 5. Conclusions

The adaptation of the Substance Abuse Attitude Survey to our cultural context and our target population of medical students has good validity and internal consistency. The questionnaire selected, beSAAS with 23 items and its three-dimensional interpretation through 2 axes: affective and cognitive, is useful and relevant to evaluate the impact of a pedagogical program on students’ representations. This study was able to highlight certain factors influencing stereotypical representations such as age, gender, personal or professional experience with substance use. The factor evaluating the interest of “Specialized treatment” clearly emerged in our study and seems to be explained by our target population and its representations influenced by the formal, informal and hidden curriculum. The study also highlighted interesting findings for improving medical education to lower stigma and provide better care for SUD people. Further studies are needed to investigate the external validity, repeatability and reproducibility of the questionnaire. 

## Figures and Tables

**Figure 1 ijerph-20-05356-f001:**
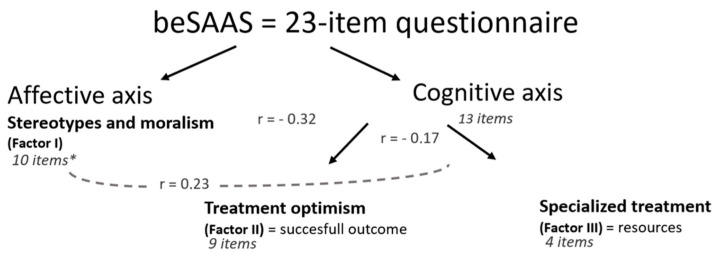
Factor structure of the beSAAS questionnaire according to the number of items characterising each factor * and correlations between scores derivated from these factors (r).

**Table 1 ijerph-20-05356-t001:** beSAAS: factors, factor loadings, Cronbach’s alpha, eigenvalues and explained variance.

Items	Factor I: Stereotypes and Moralism	Factor II: Treatment Optimism	Factor III: Specialized Treatment
		Factor Loadings	
Drug addiction is associated with a weak will	0.60		
A drug-dependent person cannot be helped until he/she has hit rock bottom	0.39		
Drug abusers should only be treated by specialists in that field			0.44
A physician who has been addicted to narcotics should not be allowed to practice medicine again	0.36		
A drug-addicted person who has relapsed several times probably cannot be treated		−0.37	
Long-term outpatient treatment is necessary for the treatment of drug addiction		0.30	
Paramedical professionals (psychologists, nurses, social workers…) can provide effective treatment for drug abusers		0.43	
Paraprofessional counselors (trained volunteers, previous drug users) can provide effective treatment for drugs abusers		0.40	
Drug addiction is a treatable illness		0.46	
Group therapy is very important in the treatment of drug addiction		0.53	
A hospital is the best place to treat a drug addict			0.51
Most drug-dependent persons are unpleasant to work with as patients	0.38		
Pregnant women who use drugs should be punished	0.76		
Coercive pressure, such as threat or punishment, is useful in getting resistant patients to accept treatment	0.48		
Alcoholism is associated with a weak will	0.65		
An alcohol- dependent person cannot be helped until he/she has hit rock bottom	0.44		
Alcohol should only be treated by specialists in that field			0.53
An alcohol-dependent person who has relapsed several times probably cannot be treated		−0.45	
Alcoholism is a treatable illness		0.53	
Group therapy is very important in the treatment of alcoholism		0.61	
A hospital is the best place to treat an alcoholic			0.56
Most alcohol-dependent persons are unpleasant to work with as patients	0.42		
Pregnant women who use alcohol should be punished	0.76		
Eigenvalues	4.0	1.7	1.3
Cronbach’s alpha	0.80	0.70	0.59
% of variance explained	40.3	17.1	12.6
% of total variance explained: 70.0			

## Data Availability

The dataset used and/or analyzed during the current study is available from the corresponding author on reasonable request.

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
