# Peer review of "Exploratory Factor Analysis of a French Adapted Version of the Substance Abuse Attitude Survey among Medical Students in Belgium"

_ijerph, 2023, doi:10.3390/ijerph20075356_

Round 1
Reviewer 1 Report
Do you think that the exclusion criteria (questionnaires with fewer than 10 responses, those with responses for only one of the two substances, and those with no socio-demographic data to be invalid) are justified? Who determined these criteria and based on what?
Please explain the following statement "Several items were highly correlated (r > 0.30)". Is it justified?
In the "Statistical methodology" section, you present certain results of the analysis. Please make a clear distinction between statistical methods and analysis results.
Please remove the "Ethics Committee" section, as it is listed at the end under "Institutional Review Board Statement"
Specify table names more clearly.
Somewhere you specify "." and somewhere "," Please uniform and correct the numbers using periods and not commas.
You talk about weak, good, and excellent correlation. Before using these terms in the results, you must clearly state the reference intervals in the methods!
Below Table 2, you must provide explanations for all superscripts.
When you mention SD in the tables, please add the prefix ±
It is awkward to use the expression "In our sample, which also seems to be representative...". You have to be exact.
You use the phrase "seems to" too much throughout the discussion. Please be more scientifically accurate and precise.
You have well noticed some limitations of your study, while you have not noticed many limitations or are not even aware of them. Please consider all limitations!
And in the conclusion, use the phrase "it seems" again. The first and last sentences of the conclusion are in contradiction.
Please remove any restrictions that you are aware of and correct the manuscript according to the above instructions and resubmit it.
Author Response
Please see the attachment.
Lou Richelle

Reviewer 2 Report
The presented work is very interesting.
Recommendations
1. Improve quality of Fig. 1
2. Add example of questionnaire to suppl materails
3. Explain more about how to improve pedagogical (educational) program by the obtained results
Author Response
Response to Reviewer 2 comments
Point 1: The presented work is very interesting.
Response 1: We thank the reviewer for this encouraging comment and his recommendations that helped us to improve the manuscript.
=====================================================
Point 2: Improve quality of Fig. 1
Response 2: We did improve it. You can see the new figure on page 6 line 193.
=====================================================
Point 3: Add example of questionnaire to suppl materials
You can find the questionnaire in Appendix A : Questionnaire bSAAS (29 items) and sociodemographic characteristics.
=====================================================
Point 4: Explain more about how to improve pedagogical (educational) program by the obtained results
We thank the reviewer for this constructive comment. Additional information has been added on page 10 line 34 :
“Lastly, it is interesting to note that the students who enrolled in the SUD training scored lower on stereotypes and moralism compared to those who did not decide to attend. Our study highlighted also an interesting result for our SUD training : specific contact with SUD people was associated with the lower score in stereotypes and moralism. Furthermore there wasn’t a statically significant difference in terms of optimism about treatment between the two groups which was a tendency to be in favor. These findings lead us to the conclusion that in this optional training, we should involve people in recovery and peer helpers to contribute to lower stigma and to help improve their perception of treatment outcomes through the sharing of positive experiences. We should also focus our teaching on recovery approach and long-term support for this chronic disease ("care vision") rather than focusing on curing the individual ("cure vision") as is commonly taught in medicine. Even if we could observe a slight improvement in stereotypes and moralism from year to year which could be explained by more awareness among medical students on mental health and substance use disorders in a context of overall vulnerability, there is still work to do. This study also reveals the interest and necessity of making a SUD training program compulsory for all medical students, in addition to the basic SUD education, to improve attitudes, access and quality of care for people with SUD [27,33,34]”
Reviewer 3 Report
1. It is not clear how factors 1 (stererotypes and moralism), 2 (treatment optimism), and 3 (specialized treatment) were chosen. Please describe deeply.
2. How was validate the questionnaire?
3. In table 1, what represent the values in the items related to factors 1, 2 and 3, and how was obtained?
4. A Cronbach´s alpha value of 0.67 is reliable?
5. Could the confinement due to covid-19 have had an impact on the results? The authors should discuss this point
6. In the legend of table 2, please add meaning of the abbreviations BE, BHE. SUD, MG, and of medical specialties. Also, include the meaning of the superscripts letters.
7. Reference 34 in line 330 is not found in the reference´s section.
Reviewer 4 Report
The article under study is interesting because of the importance of knowing how medical students, even when they are aware of the effects of consuming restricted or prohibited substances.
The introduction could be improved and mention some reasons why students resort to the use of prohibited substances.
In the results they use tables and only one figure; they could use some diagrams or graphs to make the results more evident and emphasize them.
As for the conclusions, the study is conclusive of some factors that may be related to the use and abuse of substances, the questions would be: is it possible to implement some treatments? how should the problem be addressed to reduce the consumption of this type of substance? How or what causes this type of behavior in students?

Author Response
Response to Reviewer 4 Comments
Point 1. The article under study is interesting because of the importance of knowing how medical students, even when they are aware of the effects of consuming restricted or prohibited substances.
Point 1.We thank the reviewer for this encouraging comment on an important topic.
=====================================================
Point 2. The introduction could be improved and mention some reasons why students resort to the use of prohibited substances.
Response 2. We tried to improve the introduction in keeping the same direction as you can read here on page 2 line 68:
“Given that an adaptation to our cultural and societal context was necessary, we named this new questionnaire the “bSAAS”( See Appendix A). We added questions about personal characteristics identified in the literature as influent on SUD representations such as gender, age, origin, experiences related with substance use and choice of specialty [15].”
=====================================================
Point 3. In the results they use tables and only one figure; they could use some diagrams or graphs to make the results more evident and emphasize them
Response 3. We thank the reviewer for these good suggestions which improved our manuscript. Instead of Table 2 in the main text we performed 3 forest plot graphs with relevant variables for each factor ( page 8 line 237).
=====================================================
Point 4. As for the conclusions, the study is conclusive of some factors that may be related to the use and abuse of substances, the questions would be: is it possible to implement some treatments? how should the problem be addressed to reduce the consumption of this type of substance? How or what causes this type of behavior in students?
Response 4. With our study, we wanted first to carry out an exploratory factor analysis of our bSAAS questionnaire (Appendix A) to ensure its good internal consistency and structure. The results of this part of the study are presented in Figure 1, Table 1, Appendix B and Table S1. Our second objective was to identify whether student’s representations were influenced by certain socio-demographic characteristics (Figures 2 to 4 and Table S2). Among the students’ characteristics, we have chosen to collect personal substances consumption and entourage consumption. Our hypothesis was that these variables could influence the students’ representations of substance use disorders and people who use drugs as we demonstrated it in our study, based on the literature (Richelle et al. 2022,reference 15). We did not perform a specific analysis of students ‘substance use, although this is a question which deserves attention due to the increased risk of developing substance use and related problems because of the inherent stress associated with the medical course. It is needed to consider the rate of 44% of substance use in respondents (32% for cannabis, 12,1% for multiple substances) with limitations as the focus was on the familiarity with illicit substances by asking if they have ever tried these substances (knowing that they were more familiar with licit substances) than assess substance use disorders in medical students. This specific analysis could be an interesting focus in future studies on SUD representations among medical students (considering the taboo around it).
Round 2
Reviewer 1 Report
Thank you for the answers. I suggest that you also add Table 1 and Figure 2 to the supplementary materials, given that the key content is contained in the text of the manuscript itself.
Author Response
Response to Reviewer 1 Comments
Point 1. Thank you for the answers. I suggest that you also add Table 1 and Figure 2 to the supplementary materials, given that the key content is contained in the text of the manuscript itself.
Response 1. We thank the reviewer for these suggestions. If the reviewer doesn’t mind we would like to keep Table 1 in the main text as it is important for us to illustrate more in details in the manuscript our exploratory factor analysis. In this way we make sure that readers understand directly our results, the final items, factors retained and the choice of their names thanks to both the text and the table. We agree to put Figure 2 in the supplementary materials (we added an Appendix D).
We also worked to improve the introduction and references as you can see highlighted in yellow in the revised manuscript.
We added this reference : “Cernasev A, Kline KM, Barenie RE, Hohmeier KC, Stewart S, Forrest-Bank SS. Healthcare Professional Students' Perspectives on Substance Use Disorders and Stigma: A Qualitative Study. Int J Environ Res Public Health. 2022 Feb 27;19(5):2776. doi: 10.3390/ijerph19052776” as a reference for qualitative research and took out the reference 13: “Muzyk AJ, Tew C, Thomas-Fannin A, Dayal S, Maeda R, Schramm-Sapyta N, Andolsek K, Holmer S. Utilizing Bloom's taxonomy to design a substance use disorders course for health professions students. Subst Abus. 2018;39(3):348-353. doi: 10.1080/08897077.2018.1436634. Epub 2018 Mar 5” to keep a reasonable number of references in total.
We hope that we answered correctly to the Reviewer’s expectations, don’t hesitate to guide us if it is not the case.